# Modeling of Time Geographical Kernel Density Function under Network Constraints

**Zhangcai Yin** [1] , **Kuan Huang** [1,*] , **Shen Ying** [2] , **Wei Huang** [1] **and Ziqiang Kang** [1]

1   School of Resources and Environmental Engineering, Wuhan University of Technology, Wuhan 430070, China; yinzhangcai@whut.edu.cn (Z.Y.); 253571@whut.edu.cn (W.H.); ziqiangkang@whut.edu.cn (Z.K.)
2   School of Resource and Environmental Sciences, Wuhan University, Wuhan 430070, China; shy@whu.edu.cn
*   Correspondence: 253463@whut.edu.cn

**Abstract:** Time geography considers that the probability of moving objects distributed in an accessible transportation network is not always uniform, and therefore the probability density function applied to quantitative time geography analysis needs to consider the actual network constraints. Existing methods construct a kernel density function under network constraints based on the principle of least effort and consider that each point of the shortest path between anchor points has the same density value. This, however, ignores the attenuation effect with the distance to the anchor point according to the first law of geography. For this reason, this article studies the kernel function framework based on the unity of the principle of least effort and the first law of geography, and it establishes a mechanism for fusing the extended traditional model with the attenuation model with the distance to the anchor point, thereby forming a kernel density function of time geography under network constraints that can approximate the theoretical prototype of the Brownian bridge and providing a theoretical basis for reducing the uncertainty of the density estimation of the transportation network space. Finally, the empirical comparison with taxi trajectory data shows that the proposed model is effective.

**Keywords:** time geography; kernel density estimation; potential network area; space–time trajectory

## 1. Introduction

Time geography considers that the possibility of moving objects at different accessible locations is not always equal, so quantitative spatiotemporal uncertainty analysis requires measuring the actual visit probability distribution [1]. A common method is to assign location probabilities to the potential path area (PPA), which is used in time geography to describe the potential range of a moving object during two anchor points [2]. In probability theory, this spatiotemporal uncertainty during the period of two anchor points is described by the Brownian bridge [3–6], whose density cloud is similar to a saddle formed by superimposing the bimodal peak on a ridge. However, the Brown bridge, which is inferred from random walks in a homogeneous space, is not suitable for heterogeneous ones, especially transportation networks with restricted directions [7]. Therefore, we expect a saddle-shaped probability distribution model consistent with the Brownian bridge in the transportation network space.

Downs proposed a time-geographic density estimation (TGDE) method based on the PPA kernel function [8] and extended it to the transportation network to be used in the estimation of missing points in travel itineraries and the evaluation of food availability [9,10]. PPA constrained by the transportation network is also called a potential network area (PNA) [11,12]. The TGDE kernel function corresponding to PNA is consistent with the principle of least effort [13]. It believes that the smaller the cost of a point $x$, the greater the weight, and the corresponding cost measurement is based on the least-cost path (denoted as LCP-$x$) passing through that point. In this way, the minimum-cost path between two anchor points (corresponding to the focal length in the geographic ellipse, referred to here as the focal line) is assigned the same and maximum density at any point, and the density of other points around

the focal line attenuates with cost. This kernel function corresponds in form to the base of the saddle-shaped Brown bridge and is therefore effective for such a PNA with multiple paths without intersections except for the endpoints.

So far, however, the PNA kernel function still only simulates the base of the saddle-shaped Brown bridge. This article develops two important extensions based on the divide-and-conquer strategy. First, the saddle-shaped Brown bridge is decomposed into a double-layered peak and ridge. This peak is consistent with the first law of geography [1,14]. It is based on the recognition that the observed track point should have a greater certainty or visit probability than the surrounding unobserved potential location points. The first law of geography is different from the principle of least effort mentioned above, and accordingly, the density of the peak is different from that of the ridge. We will develop here the mathematical foundations to calculate the peak density that decays with the cost distance to the anchor point. Second, in the process of our development of these foundations, we also expand the structural framework of the PNA kernel function by introducing the peak model on the basis of the ridge model, now being able to build a saddle-shaped kernel function on the transportation network. As a result, we will have a complete theory for the kernel density function in PNA space.

The other parts of this article are organized as follows: Section 2 presents the related concepts and density estimation of time geography and provides the relevant background of this article. Section 3 describes in detail the kernel density function modeling method for PNA, including the density modeling of the upper layer in accordance with the least-cost economic law [13] and the density modeling of the lower layer in accordance with the first law of geography. Section 4 describes the research process of this article, that is, using the model proposed in Section 3 to generate a kernel function of the actual PNA and comparing and analyzing it with the empirical model and the model before improvement. Section 5 summarizes the research in this article and discusses potential extensions.

## 2. Research Background

The space–time uncertainty measurement between the space–time locus points has qualitative time-geographical methods and quantitative kernel density evaluation (KDE) methods. The former provide the potential scope of activities but do not distinguish internal differences, while the latter distinguish internal differences but have a scope of action lacking a clear physical meaning. Both have become essential tools for GIS analysis and also an important basis for the model proposed in this article.

### 2.1. PNA Measurement

Time geography provides an important theoretical framework for measuring the continuous space–time uncertainty of moving objects during discrete space–time trajectory points [15]. One of its key concepts is the geographic ellipse, PPA, which represents the reachable range of a moving object under the constraints of two spatiotemporal trajectory points. Mathematically, the sum of the minimum time costs for each point in the PPA to reach the two anchor points does not exceed the time budget between the two anchor points. The same applies to PNA:

$$\text{PNA} = \left\{ x \mid t_p(s,x) + t_p(x,e) \leq t(s,e) - t_a(s,e) \right\} \tag{1}$$

where $t(s,e)$ is the time budget from $s$ to $e$, during which the stationary activity time is recorded as $t_a(s,e)$; $t_p(s,x)$ and $t_p(x,e)$ are the two minimum time costs from an anchor point $s$ to a certain point $x$ and $x$ to another anchor point $e$, respectively. For example, the height of the green line in Figure 1 at $e$ means $t_p(s,x_2) + t_p(x_2,e)$. In addition, the right formula represents the maximum time budget for movement, i.e., the minimum time cost for an object to pass through the distance of the long axis (e.g., the path $s$-$x_3$-$e$), corresponding to the height of the blue line in Figure 1 at the point $e(2a)$; the minimum value of the left formula (denoted as $t_p(s,e)$) represents the minimum time cost for an

object to pass through the focal length (e.g., the path $s$-$x_1$-$e$), corresponding to the height of the red line in Figure 1 at the point $e(2c)$.

**Figure 1.** A PNA derived from a pedestrian network based on 8 neighborhoods in a homogeneous space and its three potential paths.

As an extended concept of time geography in the transportation network, PNA has been applied in the fields of geography, ecology and transportation [16] in studies such as accessibility analysis and planning [17–19], traffic flow simulation and optimization [20,21] and behavior patterns and dynamic interaction analysis [22–25]. However, PNA only describes the location range and does not distinguish the access probability of each location. From the perspective of probability theory, PNA corresponds to the sample space, and each position point corresponds to a sample point; this provides a theoretical basis for the construction of the kernel density function on PNA.

### 2.2. Probabilistic PNA

Generally, there are two methods for constructing density clouds distributed on PNA: one is to apply classic probability models, such as the aforementioned Brown bridge [3,5] and Markov techniques [7], and the other is to apply attenuation functions, such as TGDE [26]. The Brown bridge has been used in the construction of probabilistic PPA. The basic principle is to map the starting and ending points of the Brown bridge to two trajectory points and integrate the normal distribution at all times (Figure 2a). This establishes a direct connection between time geography and probability theory and also provides a prototype and reference model from the field of probability theory for the study of time and space uncertainty in time geography, that is, saddle double-layer architecture = ridge + peak (Figure 2b). The planar PPA can be expressed as a mesh PNA in the GIS, which means that the probabilistic PPA inferred by the Brown bridge can be converted into a probabilistic PNA. However, the Brown bridge assumes that the space is homogeneous and isotropic, which is harsh for a nonhomogeneous transportation network, so the PNA density distribution needs to be reconsidered.

Another probabilistic PNA was proposed [26], and its attenuation kernel function instead of Brown bridge can be described as follows:

$$p_t(x) = PPT^* \left( \frac{t_p(s,x) + t_p(x,e)}{t(s,e) - t_a(s,e)} \right) \tag{2}$$

where $p_t(x)$ is the density of any point $x$ within the PNA; $PPT^*()$ is a distance attenuation function. The numerator and denominator respectively correspond to the left and right forms in Equation (1), so the value of the fraction does not exceed 1. For the focal length line with the smallest time cost, since the fractional value of each point is the same and

the smallest, the density of each point (including two anchor points) is the same and the largest in the PNA. This also means that other points that deviate from the focal line have a smaller density due to increased cost.

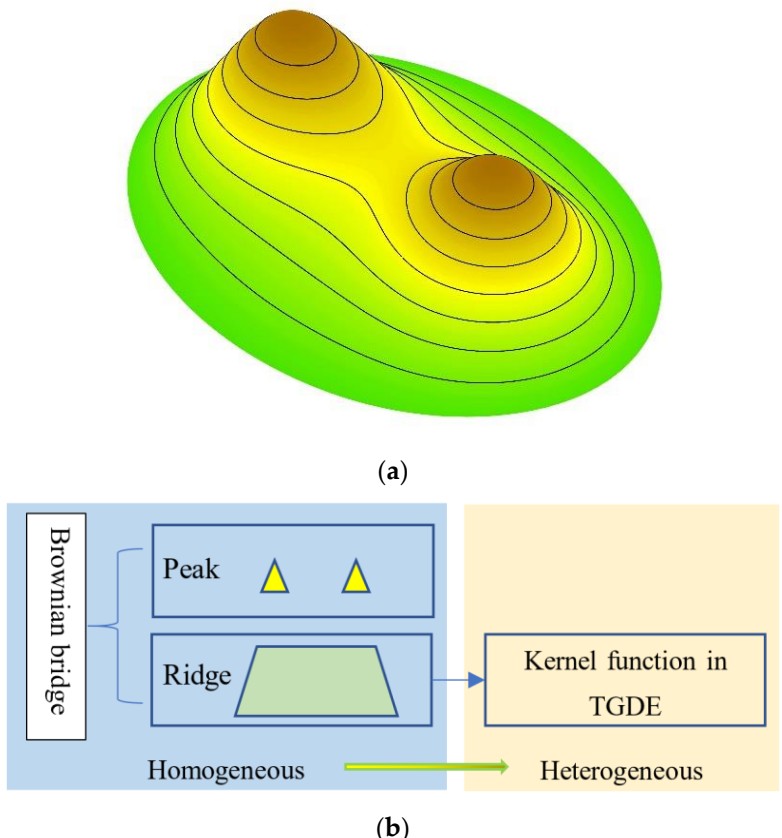

(**a**)

(**b**)

**Figure 2.** (**a**) Schematic diagram of probabilistic PPA [3] and (**b**) its two-layer architecture.

In morphology, the density cloud of $p_t(x)$ is a kind of "ridge" centered on the focal length line and attenuated to the surroundings, corresponding to a saddle-shaped base (Figure 2b). Compared with the complete structure of the ideal Brown bridge, the kernel function of the actual PNA requires the construction of the "peak". Its theoretical cornerstone comes from the first law of geography, and its physical meaning is that the track points based on observation are more certain than the potential points based on the interpolation between the track points.

### 3. Kernel Density Function in Time Geography

The main purpose of this paper is to realize that the kernel function on PNA has a saddle shape which is close to the ideal Brown bridge. Considering that the classical KDE and TGDE correspond to the "peak" and "ridge" of the saddle structure, respectively, this paper adopts a divide-and-conquer strategy to construct this saddle density function with a hierarchical structure. The basic idea is to first construct the PNA's "ridge" and "peak" density distributions independently and then compound them into a saddle shape (Figure 3). The kernel density function is essentially a decay function, and the "ridge" and "peak" are no exception. They are mainly different in attenuation center: the former is the focal length line between two anchor points, and the latter is the two anchor points. This strategy not only makes full use of existing models, but also realizes their complementary advantages through integration.

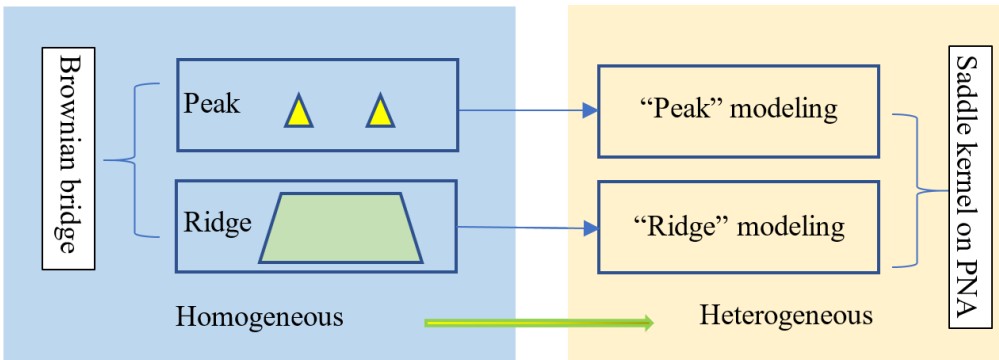

**Figure 3.** The mapping between the saddle shape of the Brownian bridge and the kernel function of the PNA.

### 3.1. Ridged Density Function

The ridge-type density function is derived from the Equation (1) of PNA, and its attenuation factor is related to the travel time budget and the least-time-cost path through the weight point, which can be formalized as follows:

$$W_{ridge}(x) = f\left(\frac{t_p(s,e|x) - t_p(s,e)}{t(s,e) - t_a(s,e) - t_p(s,e)}\right) \tag{3}$$

where the weight of point $x$, $W_{ridge}(x)$, is a decay function $f$ (*); $t_p(s,e|x)$ represents the time cost of the least-cost path passing point $x$. The above equation can also be written as $W_{ridge}(x) = f\left(\frac{t_p(s,e|x) - 2c}{2a - 2c}\right)$, where $2a$ and $2c$ are the long axis and focal length of the PNA, respectively. The density cloud of $W_{ridge}(x)$ is consistent in shape with the kernel function of TGDE (Equation (2)), which can be called a ridge (Figure 4a). In a homogeneous space, the boundary of the PNA is an ellipse; the isodensity line of $W_{ridge}(x)$ is a sequence of elliptic lines with the same focus and different elevations, due to the equal $t_p(s,e|x)$ at any point $x$ on the same elliptic line.

Taking time as a measure, the numerator is $[0, 2a - 2c]$ and the denominator is a constant $2a - 2c$, so the value range of the fraction is $[0, 1]$. Note that $2c$ is subtracted from both the numerator and denominator, which is one of the differences between Equations (3) and (2). The purpose is to standardize the fraction range from $[c/a, 1]$ to $[0, 1]$. Since the inverse distance weights (IDWs) of the fractional values {0 (at the focal length line), $c/a$, 1 (at the boundary)} are {max, middle, min}, Equation (3) can expand the interval length and weaken the boundary effect. Another difference is that the cost measure of path LCP-$x$ uses $t_p(s,e|x)$ instead of $t_p(s,x) + t_p(x,e)$. Since the former considers the cost of turning at point $x$ and the latter does not, the two are not always equal. For example, in Figure 5, the PNA is composed of three road sections $R_1$, $R_2$ and $R_3$. Their minimum costs are 11, 12 and 13, respectively. The cost of turning right at $x$ from $R_1$ to $R_3$ is 2, and the cost of going straight at $x$ from $R_2$ to $R_3$ is 0. Then, $t_p(s,e|x) = R_2(12) + \text{straight}(0) + R_3(13) = 25$, while $t_p(s,x) + t_p(x,e) = R_1(11) + R_3(13) = 24$ and $R_1(11) + \text{right}(2) + R_3(13) = 26$. Therefore, the replacement of $t_p(s,x) + t_p(x,e)$ in Equation (3) by $t_p(s,e|x)$ can improve the accuracy of cost measurement and accordingly improve the density estimation.

### 3.2. Peak-Type Density Function

The peak-type density function is derived from the classic kernel function (such as normal kernel function, triangular kernel function), and its attenuation factor is the distance from the weight point to the anchor point. It can be described as follows:

$$W_{peak}(x) = f\left(\frac{t_p(s,x)}{[t(s,e) - t_a(s,e) + t_p(s,e)]/2}\right) + f\left(\frac{t_p(x,e)}{[t(s,e) - t_a(s,e) + t_p(s,e)]/2}\right) \tag{4}$$

where $W_{peak}(x)$ is the superposition of two attenuation functions, and their two distance factors are point $x$ to the start and end anchor points, respectively; $t_p(s, x) \in [0, [t(s, e) - t_a(s, e) + t_p(s, e)]/2]$, and the fractional value range is $[0, 1]$. The above equation can also be written as $W_{peak}(x) = f\left(\frac{t_p(s,x)}{[2a+2c]/2}\right) + f\left(\frac{t_p(x,e)}{[2a+2c]/2}\right)$, which can be regarded as a linear superposition of two classical nuclear densities in form.

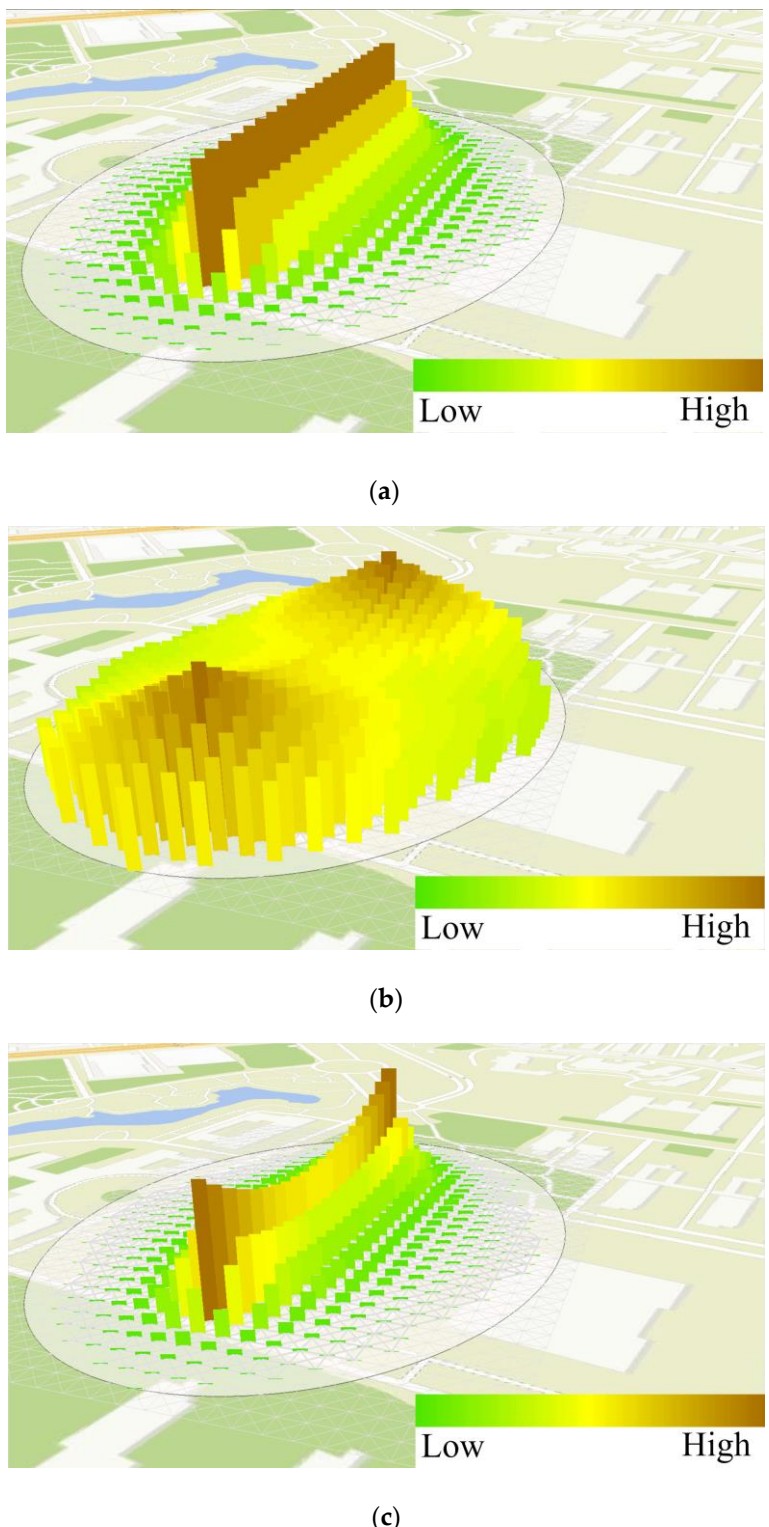

(**a**)

(**b**)

(**c**)

**Figure 4.** Saddle-shaped kernel function (**c**) composed of "ridge" (**a**) and "peak" (**b**).

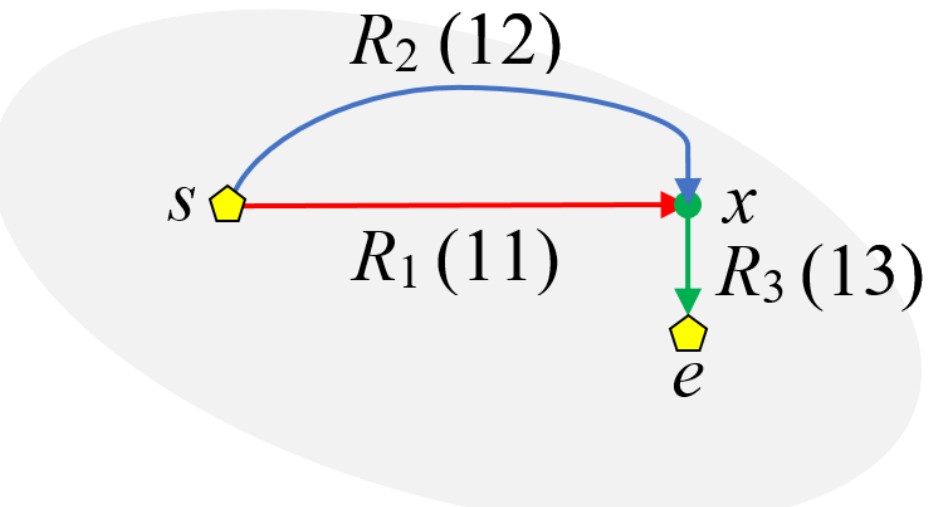

**Figure 5.** Schematic diagram of a PNA.

In terms of morphology, since the two anchor points are the attenuation centers of their respective neighborhoods, the peak density distribution will independently form two maxima at the two anchor points (Figure 4b). The shape of the two peaks has a theoretical basis from the first law of geography and a clear physical meaning. For example, the flow of traffic has the effect of diversion and confluence at the starting anchor point (source point) and ending anchor point (meeting point), respectively. This undulating form is also applicable to the focal line; that is, the density of the point in the middle of the focal line is different from (less than) that of the anchor point, which is different from the density of the "ridge".

"Peak" and "ridge" are also related in form. Equation (4) can be regarded as a separation of Equation (3); that is, one IDW based on double anchors in Equation (3) is separated into two sub-IDWs based on single anchors in Equation (4). Each sub-IDW uses a single anchor point as the center to assign a weight that decays with the distance from the center to the weight point. The above connection also means that "peak" and "ridge" have no clear boundary between them and can be transformed into each other under certain conditions, such as when the two anchor points are colocated.

*3.3. Saddle-Shaped Kernel Density Function*

The "peak" and "ridge" models have their own rationality. For the "ridge", under the condition of limited time and resources, most people will choose the least-cost path (economic law), while a few people will choose other paths for various reasons, such as avoiding congestion or passing fewer traffic lights. For the "peak", the closer to the observation point, the greater the certainty of geographic events according to the first law of geography. The rationality of each exhibits certain limitations and one-sidedness due to the failure to integrate under a unified framework. For the midpoint of the focal length line, for example, "ridge" gives the same maximum density as the anchor point, while "peak" gives a value less than the density at the anchor point. Therefore, it is necessary to integrate the "peak" and "ridge" models on the same PNA in order to take into account the rationality of each and give full play to the overall advantages.

Here, an integration method based on dot multiplication is adopted:

$$p_x = \frac{W_{peak}(x) \cdot W_{ridge}(x)}{\sum_{x \in PPN} W_{peak}(x) \cdot W_{ridge}(x)} \tag{5}$$

where $p_x$ is the PNA kernel function composed of "peak" and "ridge", which has a saddle-shaped characteristic (Figure 4c). Since "peaks" and "ridges" are based on different principles from two different disciplines, namely the principle of least effort in economics and

the first law of geography, they can be reasonably assumed to be independent. Note that the components of the kernel function that are constrained by time geography include not only the domain of definition (PNA), but also the analytical equation. In addition, because the PNA kernel function has the characteristics of a probability density function (the cumulative density is 1), there is no need to set a coefficient that responds to a variable network structure, such as the dimensionality correction coefficient of Downs and Horner [26].

The negative exponential decay model, which has been applied to potential accessibility assessment and access probability simulation [13,27], is also used in this article:

$$f(t) = \exp(-\beta t), \ \beta > 0, t \in [0, \ 1] \tag{6}$$

where the $\beta$ coefficient affects the degree of attenuation (as shown in Figure 6a): the smaller the $\beta$, the smoother. Under the condition of the same $\beta$ coefficient, the weight functions of two different intervals [0, 1] and [2/3, 1] (Figure 6b) are normalized to generate the probability density function (Figure 6c). The boundary effect of the interval [2/3, 1] is more significant than that of the interval [0, 1]. This also explains the reason why the numerator and denominator of Equation (3) increase the $-2c$ term to make the fraction interval [0, 1].

There is no uniform standard for setting the $\beta$ value in the negative exponential model [13]. Mathematically, $W_{peak}(x)$ can be regarded as the zoom factor of $W_{ridge}(x)$ in Equation (5), which adjusts the height difference between the top of the peak and the bottom of the saddle. The height difference is positively correlated with the $\beta$ value in $W_{peak}(x)$. When the $\beta$ value is 0, the proposed saddle shape degenerates into a "ridge"; this means that the traditional "ridge" model is a special case of the proposed model. In addition, the difference in the utility of the economic law and the first law in specific applications also affects the magnitude of the two $\beta$ values.

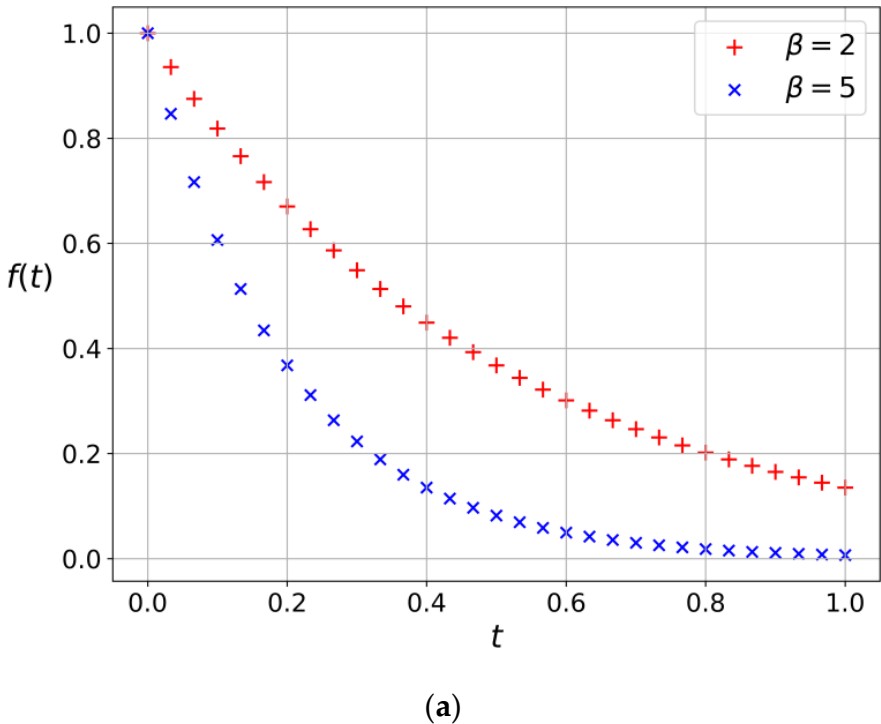

(**a**)

**Figure 6.** *Cont.*

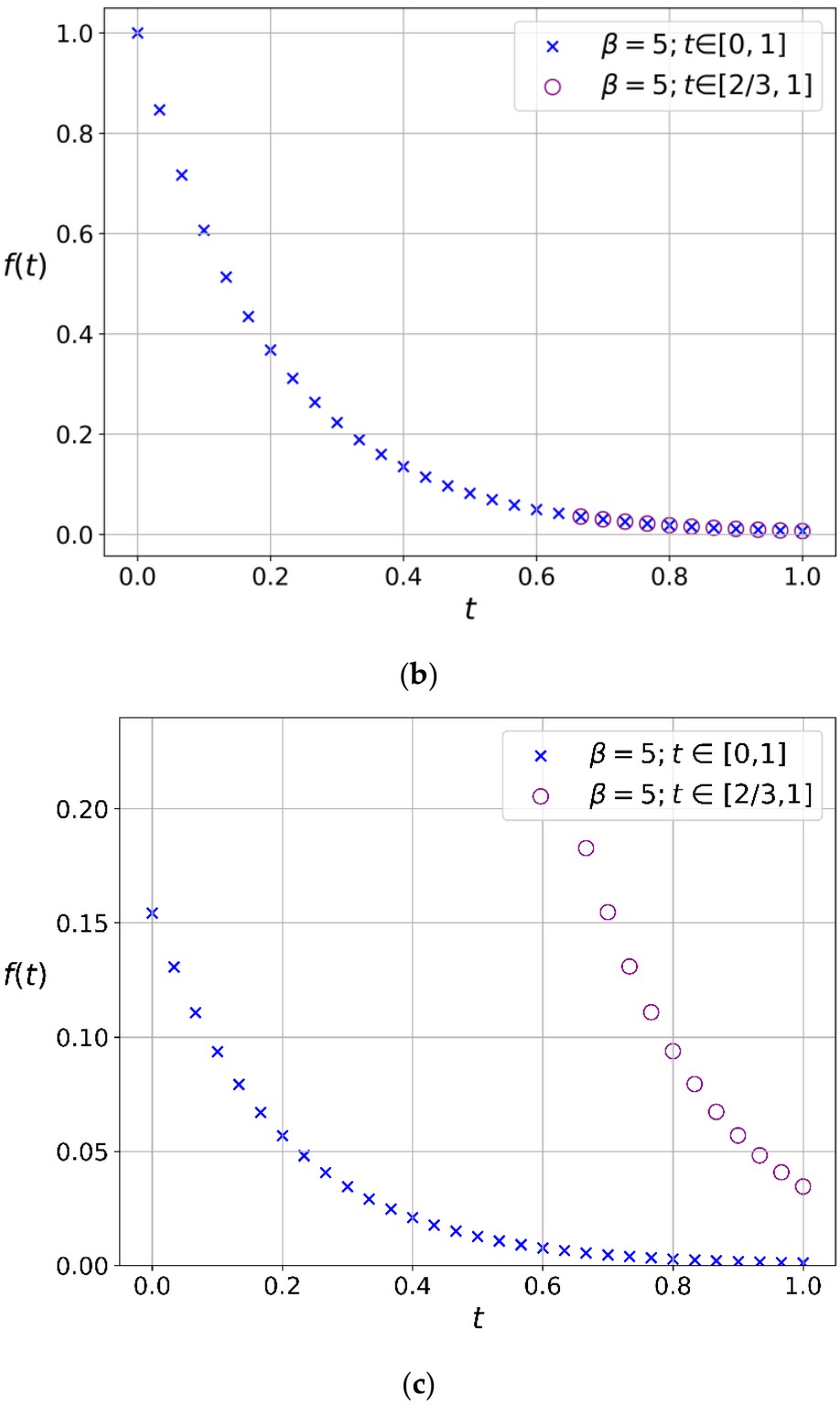

**(b)**

**(c)**

**Figure 6.** Attenuation function: (**a**) different $\beta$ values; (**b**) different intervals [0, 1] and [2/3, 1]; (**c**) normalization.

## 4. Application

### 4.1. Methods

To illustrate, the PNA kernel function proposed in this article will be applied to an actual transportation network. The study area is located in the second ring road of Xi'an, China (Figure 7a), and the transportation network involved is composed of 611 nodes and

1015 road sections. The data are mainly derived from the "GAIA" Open Dataset of Didi Travel, an online car-hailing service provider (https://outreach.didichuxing.com/research/opendata/ (accessed on 10 January 2022)), including arterial roads and their (average) driving speeds and more than 2 million driving trajectories that have been desensitized. The time frame of the study is from 17:00 to 20:00 every day during the period from 8–13 October 2018. This time range was selected because the trajectory data during this period were sufficient, and the speed distribution on the transportation network was relatively stable due to the simple evening peak. In addition, the trajectory data of the positioning offset were filtered before the analysis.

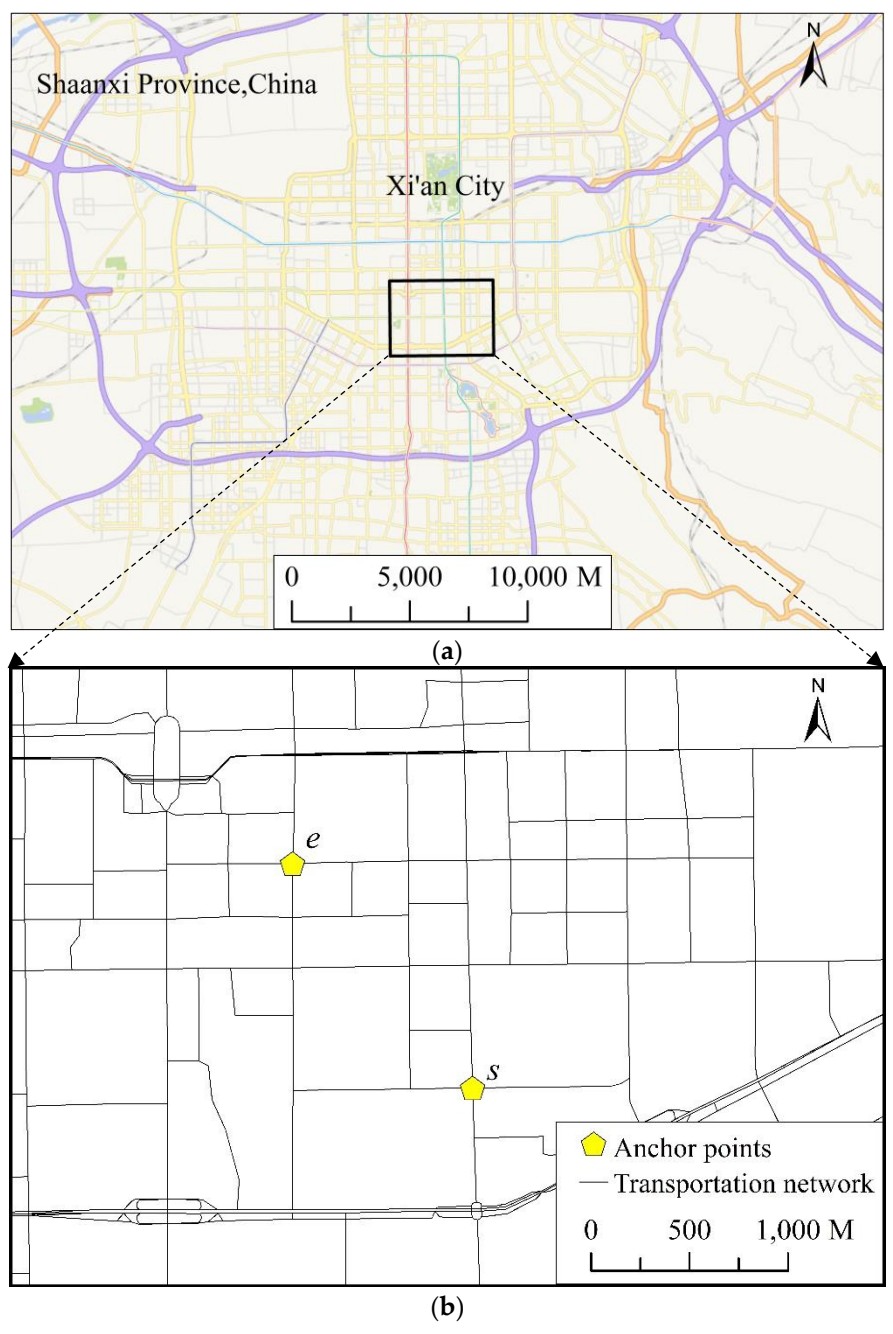

**Figure 7.** Study area (**a**) and a pair of anchor points in the transportation network (**b**).

PNA is constructed for one pair of anchor points ($s$ (108.960000° E, 34.230000° N), $e$ (108.950000° E, 34.240000° N)) using Equation (1), with a fixed activity time of 0 and a time budget of 1.5 times the minimum cost (Figure 7b). The reason for setting the value to 0 is

that taxis are usually point-to-point commuting, and the only known space–time trajectory data do not include activity time. Turning cost is a part of the transit cost based on time measurement. According to Song et al. [7], it can be set to 3 s for right turns, 15 s for left turns and 40 s for U-turns. The other part is the cost of passing roads, and its calculation needs to set the corresponding maximum traffic speed: the main road is generally 20–55 km/h according to the "GAIA" plan, and the general road is 20 km/h. The time budget, according to Papinski and Scott [28], is set to 1.5 times the cost of the focal length to allow moving objects to be uncertain because they can deviate from the focal line. The calculation of transit cost applies the extended A* algorithm to the weighted highway network [29], and the display of PNA uses ArcGIS v.10.7 (ESRI, Inc. Redlands, CA, USA).

*4.2. Results*

Figure 8 illustrates a PNA in the study area, where the thin line represents the transportation network and the thick line represents the PNA. In the thick line, the dark black line represents the focal length line, and the light black line represents other potential paths. As the domain of the kernel function, PNA determines the position and shape of the kernel density cloud: the two anchor points in the PNA determine the two maximum "peaks" in the cloud, and the focal length line in the PNA determines the "ridge" in the cloud.

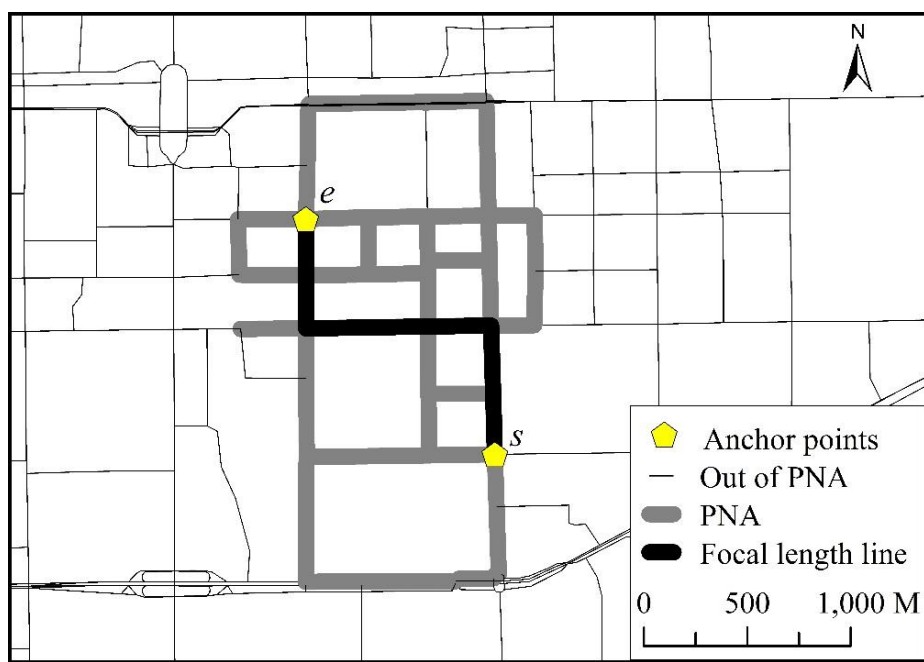

**Figure 8.** A PNA constructed by a pair of anchor points.

Figure 9 illustrates the kernel function distributed on the PNA, indicating that the higher the height, the greater the probability. Figure 9a,b illustrate the distribution of the kernel function on the node and on the transportation network, respectively. The latter based on continuous distribution is a linear fitting of the former based on discrete distribution. They show the three-dimensional characteristics of the kernel function: "peaks" (high at the anchor and low around it) and "ridges" (high at the focal line and low on both sides). Figure 9c,d show two views of the kernel function from the perspective of dimensionality reduction: the front view, with a projection plane with the straight line s-e as the horizontal axis and probability as the vertical axis, highlighting the saddle shape; a side view, with a projection plane perpendicular to the straight line s-e, highlighting the attenuation characteristics.

### 4.3. Verification

In order to test the validity of the proposed model, an empirical model is generated by calculating the frequency of the actual trajectory from the data source through the pair of anchor points. Subsequently, the coefficient of determination is used to evaluate how close the proposed theoretical distribution is to the empirical model: $R^2 = 1 - \sum(p_x - p'_x)^2 / \sum(p_x - \overline{p_x})^2$, where $p_x$ and $p'_x$ are the theoretical and actual values at point $x$, respectively, and $\overline{p_x}$ is the mean value of $p_x$. Note that the point $x$ in the equation is located on the actual trajectory within the PNA as per Song et al. [7]; when $R^2$ is a positive value, the larger the value, the better the fit. In addition, we also calculated the $R^2$ of the empirical distribution and the simple "ridge" distribution to show the improvement of the double-layer structure compared to the single-layer structure.

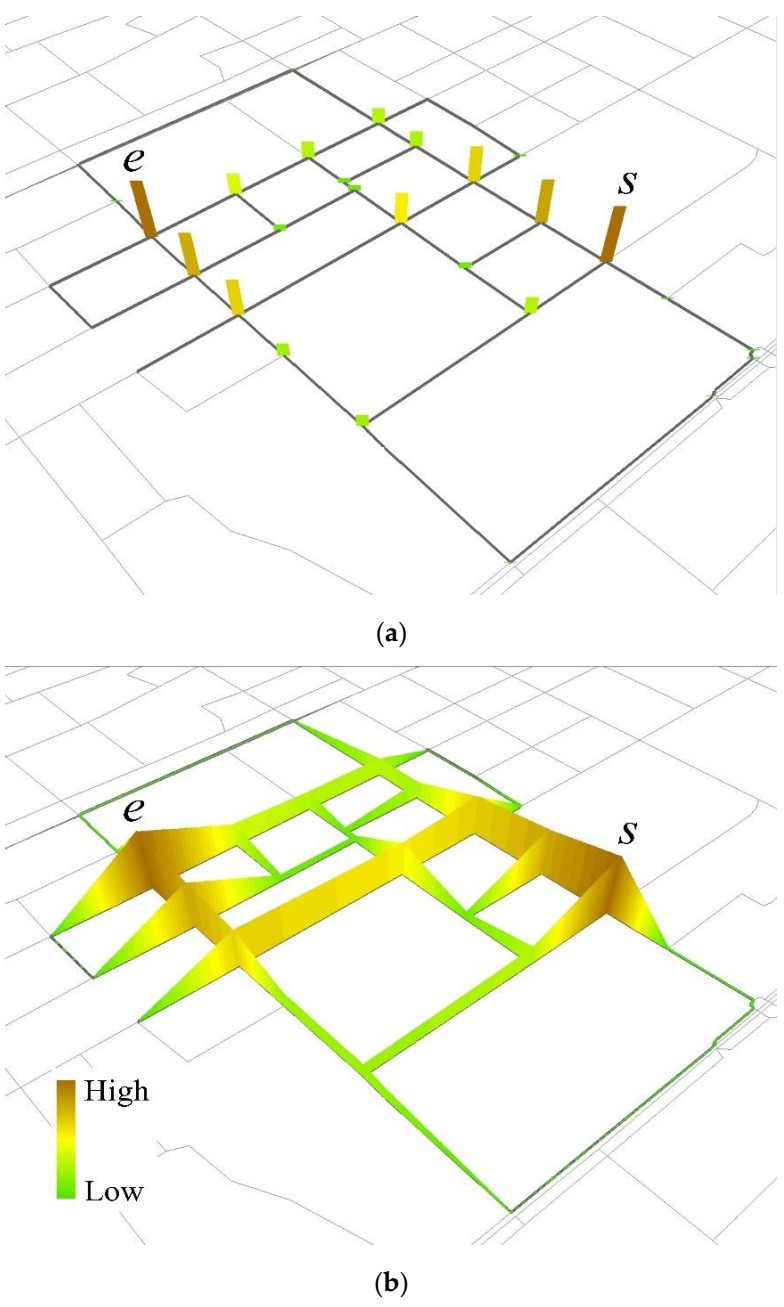

(**a**)

(**b**)

**Figure 9.** *Cont.*

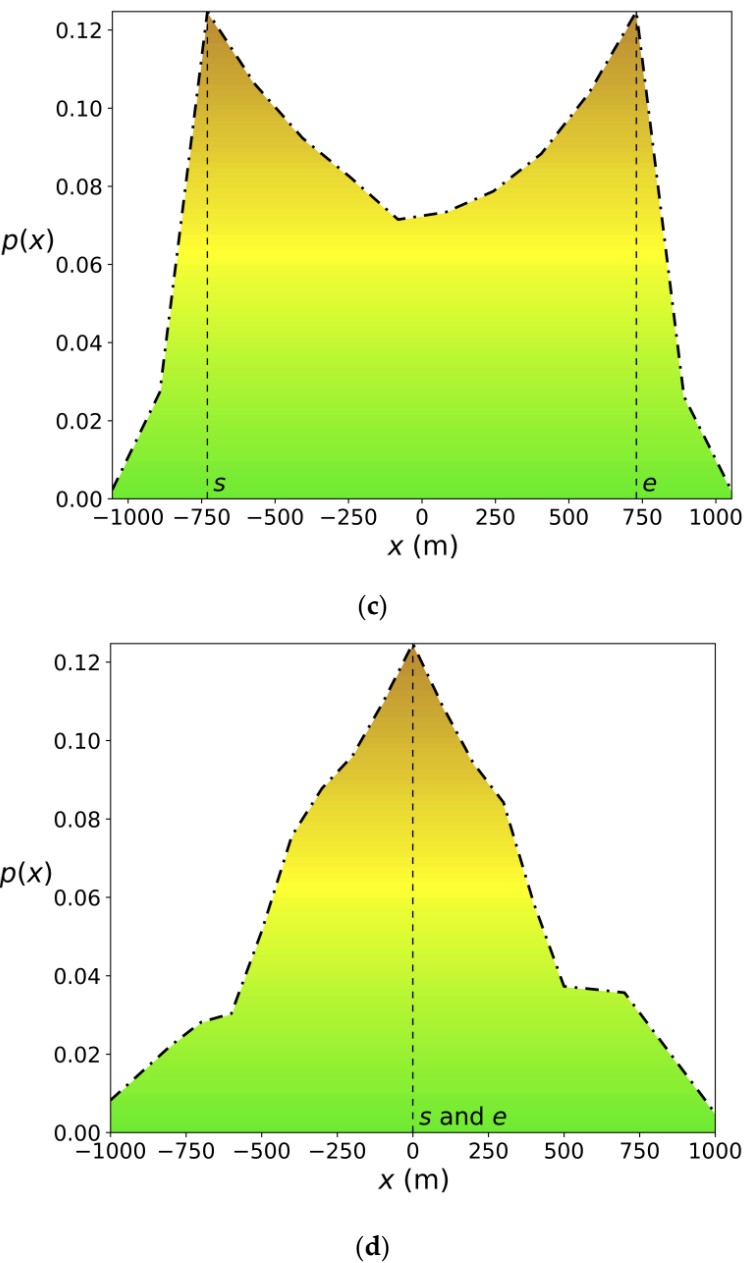

**Figure 9.** Kernel function on PNA: (**a**) discrete type; (**b**) continuous type; (**c**) front view; (**d**) side view.

Figure 10 illustrates the actual distribution of 154 trajectories on the PNA, including three-dimensional discrete and continuous density clouds (Figure 10a,b) and two-dimensional views (Figure 10c,d). It can be seen from Figures 9 and 10 that the theoretical and actual distributions are similar in both two dimensions and three dimensions, $R^2 = 0.941$. This similarity also validates the method of setting two $\beta$ values in the PNA kernel function. For the $\beta$ value in the "ridge", it cannot be set too small (the smaller the value, the flatter the density curve), in order to avoid the overestimation at the boundary caused by the too-small $\beta$ value. Dominated by economic laws, people who move directionally between paired anchor points usually do not deliberately detour beyond the PNA boundary, which is the physical meaning of the low or even zero probability of boundary. For the $\beta$ value in the "peak", it should not be set too large (the larger, the steeper), to avoid the underestimation of the saddle caused by the excessive $\beta$ value. Subject to the first law of geography, the saddle (which has a higher density due to its location on the "ridge") is not higher than the anchor point. This is the theoretical basis for the greater visit probability of the saddle. In addition, we also

measured the $R^2$ between the single-layer "ridge" model and the actual distribution, 0.833. The falling value indicates the necessity of superimposing the "peak".

In addition to the above-mentioned overall perspective, the PNA kernel function can also be tested from a single perspective of the path. Figure 11 shows the three distributions of the proposed model, the "ridge" model, and the empirical model for each of the red, green, and blue paths in Figure 10a. In terms of cost, the red line is the smallest and the blue line is the largest. It can be seen from the figure that the proposed model is closer to the empirical model than the "ridge" model: the red line of the two-tuple $R^2$ between the proposed model and the empirical model, $R^2$ between the "ridge" model and the empirical model is (0.718, −0.531); the green line, (0.927, 0.624); and the blue line, (0.842, 0.454). This also explains the statistical significance that the PNA kernel function must include the "peak".

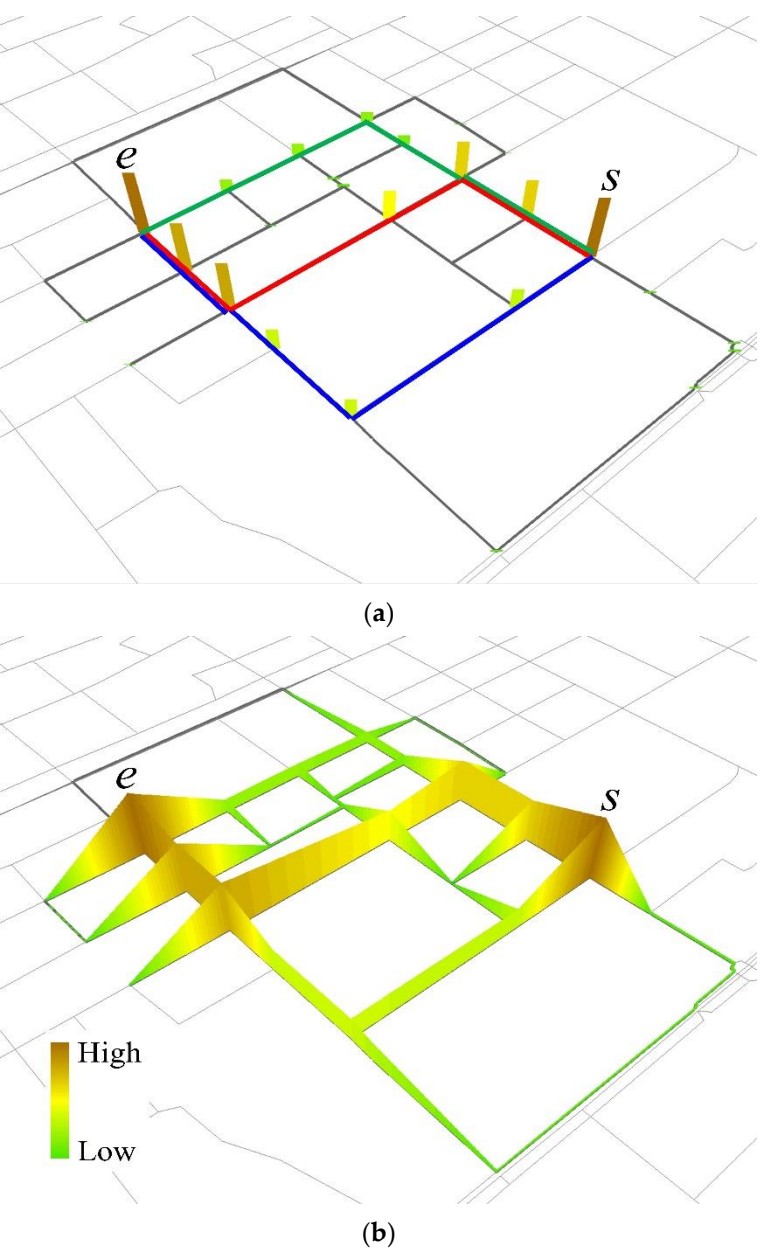

(a)

(b)

**Figure 10.** *Cont.*

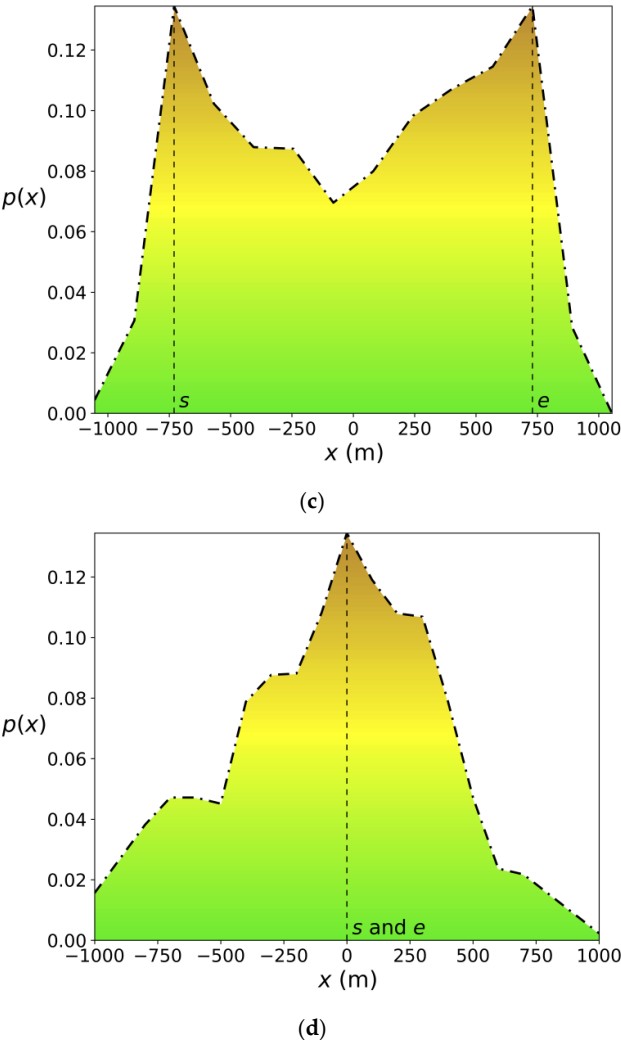

**Figure 10.** Empirical model on PNA: (**a**) discrete type; (**b**) continuous type; (**c**) front view; (**d**) side view.

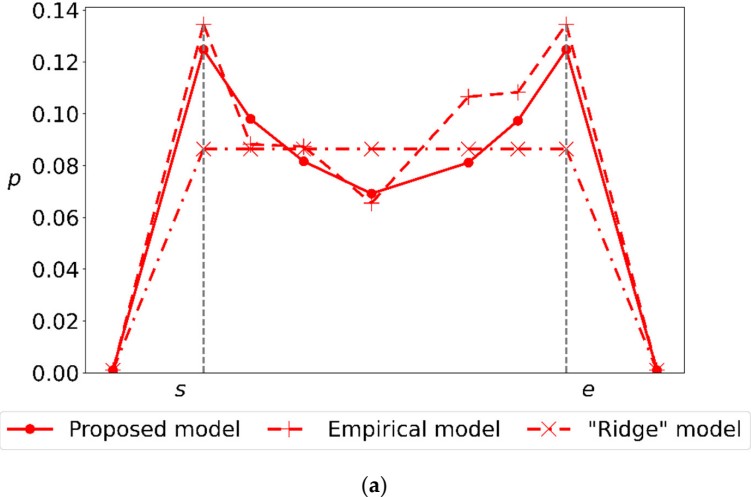

(**a**)

**Figure 11.** *Cont.*

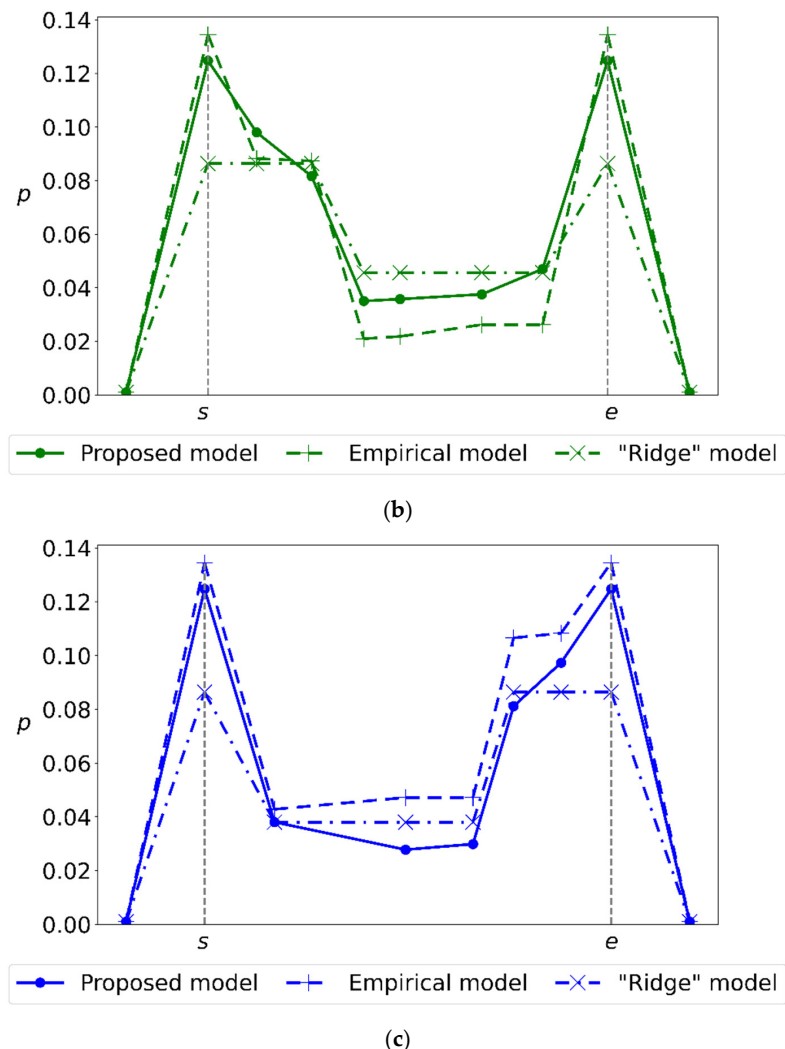

**Figure 11.** The proposed model, "ridge" model and empirical model for each path: (**a**) red line, (**b**) green line and (**c**) blue line.

## 5. Conclusions and Discussion

This paper extends a previous work on the single-layer "ridge" model of the PNA kernel function [26]. It develops the mathematical foundation for the double-layer "ridge" + "peak" model of the PNA kernel function and takes into account the economic law with the least cost and the first law of geography. It shows that the pure fact that a moving object is continuously observed in two geographic scenes breaks the previous assumption that the kernel density function is only restricted by economic laws. Therefore, the two laws of economy and geography must be introduced in the kernel density function in order to cooperatively assign locational probabilities. The two laws not only lead to the double-layer structure of the kernel density function, but also lead to a saddle shape that approximates the Brown bridge and the actual distribution. The formalization of this model is no longer just a function of constant coefficients, but a framework that can adjust coefficients ($\beta$) according to specific applications. With the Brown bridge model [3] and the two-layer model at hand, the spatial types (homogeneous and heterogeneous) covered by the time-geographic PPA kernel function are complete.

This model has been successfully implemented in Python and ArcGIS, which are used for geographic computing and geographic visualization, respectively. In addition, the resulting values of the model have saddle characteristics similar to Brownian bridges and can now be used for kernel density estimation of transportation network space, providing

a basis for the prediction of visiting intensity and dynamic interactions of moving objects. When the space–time trajectory points of a moving object on the transportation network are known, its probability of visiting any location at any point in time can be calculated. The estimated density value can be further used to analyze the meeting probability of two moving objects at any moment and the dynamic interaction within a period of time. The development of related algorithms and queries is part of future work.

Another future work is to generalize the proposed model into a planar area. Since the plane area can be expressed as a network structure through nodes and edges respectively representing grids and their neighboring relationships, the proposed model can theoretically be extended to a nonhomogeneous continuous surface space. Considering that the real space is composed of the network and the area, another extension of the presented model is required for composite spaces, such as urban space.

The model proposed in this article tends to be theoretical, and future work obviously needs to discuss its actual value or its limitations in actual scenarios. Globally deterministic and locally random spatial activities, for example, are suitable for this theory. In reality, the activities of moving objects (such as humans and animals) are random in the local area and regular in the overall situation [1], which is related to the variability of the geographical environment and the periodicity of the life course. On a certain transportation network, the frequency statistics of the periodic trajectory flow of an object or the trajectory flow of multiple objects in the same time period can form the probability density function on the PNA. On the other hand, we can estimate the probability of a mobile object visiting each location based on the limited track points and realize the spatiotemporal interpolation with probability between the track points for those applications that try to seek continuous footprints. For example, according to the traffic flow data of limited observation points, the traffic volume distribution on the whole transportation network is estimated to evaluate vehicle exhaust emission data over time [17] and support applications such as traffic flow simulation and quantitative optimization, accessibility analysis and planning. As another example, when COVID-19 is still prevalent, the access probability of the activity space is evaluated based on the limited trajectory points of the cases, which assists in the epidemic risk assessment and epidemic prevention and control in areas with no footprint records [30]. These applications will serve as examples of our model and provide a basis for the localization of the model in specific applications, such as the setting of $\beta$ parameters for specific applications.

The ideal Brown bridge provides an approximation of the PNA kernel density function. This approximation provides a theoretical basis for the comparison of different PNA kernel density functions and the $\beta$ parameter setting of the presented model. A feasible method for the latter is as follows: first, construct a dense enough transportation network to approximate the continuous homogeneous space assumed by Brownian motion; then, analyze the degree to which the PNA kernel density function approximates the Brownian bridge, and thus clarify the relationship between the $\beta$ parameters and the spatiotemporal information of the two anchor points.

Although more work needs to be done to empirically test the results across a wide range of fields and applications, it is obvious that (a) the kernel density function for directional movement involves the laws of economics for movement and the laws of geography for its scene; (b) time geography and KDE provide effective and computationally feasible methods to express this density, further supporting the analysis of movement uncertainty; and (c) some empirical studies seem to provide results that support the two-tier model.

**Author Contributions:** Conceptualization, Zhangcai Yin, Kuan Huang and Shen Ying; data curation, Wei Huang and Ziqiang Kang; funding acquisition, Zhangcai Yin; methodology, Zhangcai Yin, Kuan Huang and Shen Ying; software, Kuan Huang, Wei Huang and Ziqiang Kang; writing—original draft, Kuan Huang; writing—review and editing, Zhangcai Yin. All authors have read and agreed to the published version of the manuscript.

**Funding:** This research has been supported by grants from the National Natural Science Foundation of China (42171415).

**Institutional Review Board Statement:** Not applicable.

**Informed Consent Statement:** Not applicable.

**Data Availability Statement:** The codes that support this study are available at GitHub at the following link: https://github.com/hk133314/Probabilistic-PNA-Calculation (accessed on 10 January 2022). The trajectory data of this study provided by DiDi Chuxing is for scientific research purposes only, and the data can be obtained by contacting DiDi Chuxing using the email address of a university/research institution in China (link: https://gaia.didichuxing.com (accessed on 10 January 2022)).

**Acknowledgments:** The authors would like to acknowledge DiDi Chuxing, which provided the high-quality and massive trajectory data for this study (data source: https://gaia.didichuxing.com (accessed on 10 January 2022)).

**Conflicts of Interest:** The authors declare no conflict of interest.

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
