# Peer review of "Modeling of Time Geographical Kernel Density Function under Network Constraints"

_ijgi, doi:10.3390/ijgi11030184_

Round 1
Reviewer 1 Report
This paper proposed a new model for calculating time geographical kernel density, and demonstrated the superiority of the proposed model through verification. Overall, the paper is logically well-organized, but the following few things need to be revised. First, the validity of the model was verified for only one case(a small urban section). However, it is difficult to generalize the superiority of the presented model only through the verification of one case. In addition to the case area shown in Figure 7, evaluation should be conducted in other sections. Second, this paper adopted the divide and conquer strategy to calculate saddle kernel density on PPN. Dividing the density function into peak and ridge is possible under the assumption that the two functions are independent of each other. It is necessary to justify the assumption of independence of the peak and ridge density functions. In addition, the rationale for choosing the strategy should be presented.
Reviewer 2 Report
This paper proposes a new framework combining the ridge and the peak models to better estimate the probability or uncertainties of movement of moving objects within a transportation network. The paper is well-written, interesting, and reads well. But some additional aspects could have been discussed more. Although the current version of the manuscript is already in quite good shape, it can be improved by addressing the issues following:
1. One of my major concerns/comments is that – what is really “new” and what is the “added value” of your proposed new framework? I think I kind of get it, but it’s a bit unclear. What is innovative/new about your proposed methods compared to the two papers below:
• Downs, J.A.; Horner, M.W. Probabilistic potential path trees for visualizing and analyzing vehicle tracking data. J. Transp. Geogr. 2012, 23, 72-80, doi:https://doi.org/10.1016/j.jtrangeo.2012.03.017.
• Song, Y.; Miller, H.J.; Zhou, X.; Proffitt, D. Modeling Visit Probabilities within Network‐Time Prisms Using Markov Techniques. 446 Geogr. Anal. 2016, 48, 18-42, doi:https://doi.org/10.1111/gean.12076.
Could you more explicitly elaborate why your research is innovative and makes contributions to the field of analytical geography in your revised manuscript?
2. In the Abstract and throughout the manuscript – I thought it would be better to say “transportation network” rather than “road network”, considering the possibility that your proposed methodology is applied to other types of transportation networks as well (for instance, public transit networks).
3. In the Introduction and throughout the manuscript – I don’t think “potential path network (PPN)” is the term commonly used in the field of analytical time geography. I think what you mean is potential network area (PNA). Please see the reference below, and consider replacing the term PPN with PNA in your revised manuscript:
• Miller, H.J., 2018. Time geography. In Handbook of Behavioral and Cognitive Geography. Edward Elgar Publishing.
4. Lines 77-79 – could you explain why time geographical methods are “qualitative” here? I thought the authors are relying on quantitative and analytical time geography and space-time prisms.
5. Line 95 – what does “the long axis 2a of the PPN” mean? Also, the use of terms such as “focal length” and “long axis” are not that clear. More kind annotations and labels in Figures will be much helpful for readers.
6. Lines 97 – what is tp(s, e)? I can’t find “tp(s, e)” from the equation (1).
7. Figure 1 – could you specify and highlight what “2a” and “2c” are in Figure 1?
8. Lines 103-104 – “As an extended concept of time geography in the road network, PPN has been applied in the fields of geography, ecology and transportation [14], such as accessibility analysis and planning [15,16]”
Here, I think citing papers [15] and [16] are not that proper selections. I reckon papers that can better support your argument are like the following. I would suggest you referring these papers when you talk about the applications of network-time prisms in the fields of accessibility and planning.
• Song, Y., Miller, H.J., Stempihar, J. and Zhou, X., 2017. Green accessibility: Estimating the environmental costs of network-time prisms for sustainable transportation planning. Journal of Transport Geography, 64, pp.109-119.
• Lee, J. and Miller, H.J., 2018. Measuring the impacts of new public transit services on space-time accessibility: An analysis of transit system redesign and new bus rapid transit in Columbus, Ohio, USA. Applied geography, 93, pp.47-63.
• Jaegal, Y. and Miller, H.J., 2020. Measuring the structural similarity of network time prisms using temporal signatures with graph indices. Transactions in GIS, 24(1), pp.3-26.
9. Lines 111-113 – what about Song and Miller (2016) “Modeling Visit Probabilities within Network‐Time Prisms Using Markov Techniques” paper? Can’t the approach of this paper be considered here?
10. Figure 2(a) – I don’t think this is probabilistic PPA. I believe it is the Brownian bridge movement model. The Figure caption from the original paper you cited is “Probability density for the fraction of time spent in different regions, constructed using the Brownian bridge movement model.” Please present a proper Figure here.
11. Lines 135-140 – two sentences that are almost identical are repeated here. Please fix.
12. Lines 178-180 – isn’t it {min, middle, max} rather than {max, middle, minimum} according to the sentence in the manuscript “For the focal length line with the smallest time cost, since the fractional value of each point is the same and the smallest”?
13. Lines 183-187 – descriptions here are very confusing and not clear. Please revise. I am not so sure the summations are correct. I am also confused with what exactly right turns and going straight are in the example Figure 5.
14. Equation 4 – again, what is tp(s, e)?
15. Could you clearly explain how t(s, e) – ta(s, e) + tp(s,e) becomes 2a + 2c in the revised manuscript?
16. Equation (5) – isn’t it right to take the Sigma operation out to the front of the fraction form?
17. 4.1. Methods – when describing the dataset, I think the authors might want to explicitly say “taxi” trajectory dataset.
18. Figure 6 – I think the y-axis of Figure 6 a), b), and c) should be consistent.
19. Lines 267-268 – What does “sufficient” mean here?
20. Lines 273-275 – what does “the only known space-time trajectory does not include activity time” mean?
21. Line 282 – Please provide a reference paper for the extended A* algorithm.
22. Lines 294 – Figure 7 -> Figure 7(b)
23. Figure 7(b) – please put a legend in Figure 7(b).
24. Please specify “s” and “e” locations in Figures 8 and 9.
25. Line 319 – why is there a symbol ‘ behind 2 in the denominator?
26. Lines 345-354 and throughout the manuscript – I think using the term “statistical model” when describing the results from the taxi trajectory data analysis can be confusing. I think a better term would be “empirical model” or “actual model” or “observed model”
27. Please provide references for “Economic law” and “the first law of geography”
28. I was wondering whether the authors can share their Python code for implementing their proposed methods with readers to promote open science.
Reviewer 3 Report
This article is relevant because it is devoted to the development of a theoretical framework for reducing the uncertainty in estimating the density of the road network space. The article title adequately reflects the content. In the abstract, the authors briefly give the article essence, describe the problem state, research methods, results. Key words correspond to the article content.
In the introduction, the authors describe the models that were used by researchers to solve a problem similar to the one posed by the author, and the article structure is given. The second section includes a literature review of research on the article topic, confirming its relevance, including unsolved problems. In the third section, the authors describe different variants of the kernel density function in temporal geography and methods for constructing it. The fourth section is devoted to describing the application of theoretical development: methods, results and their verification are presented. In the "Conclusions and discussions" section, the authors summarize the findings and results and indicate directions for future research.
Theoretical and practical conclusions are supported by figures and tables, which are of adequate quality. The list of literary sources is sufficient, but can be extended.
In our opinion, the article corresponds to the topic “improving the sustainability and efficiency of the road network” and corresponds in type to the preliminary study.
Comment.
- In our opinion, it is necessary to more clearly formulate the goal and problems of the study, as well as to confirm how the methods used allow us to assess the adequacy of the conclusions given in the article.
- The list of reference, in our opinion, can be expanded, since the authors cited few studies performed in recent years and reflecting the use of new methods, models and tools for information mining to solve the problem formulated as a research topic.

Author Response
Please see the attachment.

This manuscript is a resubmission of an earlier submission. The following is a list of the peer review reports and author responses from that submission.